# Households' poverty and inequality after the COVID-19: Insights from panel data of face-to-face surveys in Southeast Asia

**Manh Hung Do●\*, Trung Thanh Nguyen●, Ulrike Grote**

Institute for Environmental Economics and World Trade, Leibniz University Hannover, Germany

\* hung@iuw.uni-hannover.de

## Abstract

The global COVID-19 pandemic has had catastrophic impacts on global economies and human health. The consequences of the COVID-19 pandemic include the fatalities of millions of people and increased poverty. Given the limited evidence at the micro level and on the heterogeneous impacts of the COVID-19, we use panel data of 2,517 households from Thailand and Vietnam to investigate the correlation and heterogeneous effects of the COVID-19 on household income and poverty and to examine the distributional effects on household income. The results show that the COVID-19 has a negative correlation with household income, while it has a positive correlation with the Gini coefficient of income inequality and poverty. Our results also show that there are unequal effects of the COVID-19 on household income, inequality, and poverty at the micro level. Further, the COVID-19 has the highest negative impact on daily per capita income of households in the 10th and 25th quantile groups. Policy implications are proposed to support households in disadvantageous groups.

## Introduction

The global COVID-19 pandemic has had catastrophic impacts on global economies and human health. The consequences of the COVID-19 pandemic include the fatalities of millions of people and increased poverty [1–3]. Amid the COVID-19 pandemic, thousands of papers have been published trying to quantify the impacts on both social and economic aspects [4]. On the one side, the impacts of the COVID-19 have been well examined at the macro level. The introduction of physical distancing measures (i.e., lock down) and the disruptions in production and supply chains caused by the COVID-19 pandemic have resulted in reduced livelihoods, increased food insecurity, increased poverty, and increased inequality [5–9]. On the other side, evidence on the impacts appears to be limited at the micro level, such as studies on rural households in developing countries, although these smallholders contribute significantly

**Data availability statement:** The data of the Thailand – Vietnam Socio-Economic Panel (TVSEP) project are available for scientific research purposes free of charge via the portal www.tvsep.de. The data and codes to produce the results of this manuscript are attached to the submission.

**Funding:** The author(s) received no specific funding for this work.

**Competing interests:** The authors have declared that no competing interests exist.

to the global food systems by producing 80% food in Asia and Africa [10–13]. Any negative impacts on these rural households can have significant implications for the development at the macro level [12,14]. Some available studies at the micro level have pointed out that the COVID-19 has led to job losses, business collapse, and livelihood losses which have further resulted in decreased income and increased poverty [1,15–21].

The unequal effects of the COVID-19 pandemic on vulnerable groups, especially on low-income populations, have further deepened the inequality of income, consumption, and economic growth since people in these groups lack of access to alternatives and resources to cope with these unexpected events [2–4,22,23]. However, studies on the heterogeneous impacts of the COVID-19 have been focusing more on the gender aspects that females were more negatively affected by the COVID-19 than males [24–27]. The question arises whether the pandemic affects rural households differently, for example, by their participation in farming, their educational level, or different income levels. Evidence on answering this question is rare.

Besides limited evidence at the micro level and on the heterogeneous impacts of the COVID-19, there are additional research gaps that need further attention. First, most of the studies at the micro level applied cross-sectional data or recalled data to assess the before-after impacts. While the former application apparently produces limited results, the latter approach might result in non-sampling errors [3]. Second, some micro studies used data collected from phone or online interviews. This research design, however, might exclude important groups of the population in developing countries such as rural households and poor people who could not afford a phone or might live in the areas with low phone connectivity or internet access. Last, the distributional impacts of the pandemic have been well captured in developed countries [19,28–30], while there are a few in developing countries. Evidence from the distributional effects might provide useful insights into understanding how the COVID-19 further deepens the issues of inequality in different income groups. Findings from our study can provide policy implications to support households in disadvantageous groups to help them mitigate extraordinary shocks such as the COVID-19 in future.

Against the above background, we examine two research questions, namely (i) the correlation and heterogeneous effects of the COVID-19 pandemic on household income, income inequality, and poverty and (ii) the distributional effects of the pandemic on household income. We contribute to the literature by addressing these research gaps. First, we employ panel data from a long-term project in two emerging economies, namely Thailand and Vietnam, which can capture the before- and after-the-pandemic periods for impact assessments. In particular, the data from the Thailand – Vietnam Socio-Economic Panel (TVSEP) project are used in this current study. Second, the data from TVSEP project have been collected from rural households in these countries which can enrich the literature by providing the evidence on the impact of the COVID-19 pandemic at the micro level. These countries are well-known for rapid economic growth, but the large share of their population live in rural areas, rely on agriculture, and are smallholder farmers [12–14]. Last, the TVSEP data have

been collected from conventional face-to-face interviews using Computer Assisted Personal Interviewing (CAPI) which can allow us to examine the impacts of the COVID-19 more accurately from a wide range of respondents than internet, phone line, or mobile phone interviews.

The TVSEP data are strongly relevant to this study because of following reasons. First, the reference period of the most recent wave of the TVSEP data collected in 2022 was from May 2021 to April 2022 which could cover the worst infection wave of the COVID-19 with the variant B.1.1.7 (Alpha) in Thailand and B.1.617.2 (Delta) in Vietnam, compared to the outbreak in 2020 [31]. The surge of the outbreak remained until the beginning of 2022. In this case, the impacts of the COVID-19 pandemic can be infected household members which can have critical implications for household income. Second, during the outbreak since April 2021, physical distancing measures were implemented in both Thailand and Vietnam to mitigate the spread of the COVID-19. In particular, lockdowns were adopted between July and September 2021 in Thailand [32] and between June and September 2021 in Vietnam [33]. These lockdowns might result in negative economic consequences [34,35]. For farming households, the negative impacts can last longer after the easy of the lockdown due to disruptions in logistics of agricultural inputs and labor shortages [36,37]. Hence, the reference period of the TVSEP data might capture the influence of the COVID-19 on rural households in the 2021 outbreak in Southeast Asia.

The reminder of this article is structured as follows. Section "*Materials and methods*" describes the study sites, data, and the descriptive statistics of the data. Section "*Research method*" explains our empirical research method. Section "*Results and discussion*" presents and discusses the results. Section "*Conclusions and policy implications*" summarizes key findings and proposes policy implications.

## Materials and methods

### Data

We rely on data from the "Thailand - Vietnam Socio-Economic Panel (TVSEP): Poverty dynamics and sustainable development: A long-term panel project in Thailand and Vietnam" funded by the German Research Foundation (DFG FOR 756/2) (see www.tvsep.de for detailed information about the project). The sampling procedure of the TVSEP followed the guidelines of the Department of Economic and Social Affairs of the United Nations [38]. The TVSEP project purposely selected three provinces in northeastern Thailand (Buriram, Nakhon Phanom, and Ubon Ratchathani) and three provinces in central Vietnam (Ha Tinh, Thua Thien Hue, and Dak Lak) as study sites. These provinces were selected because they fulfilled four key criteria for assessing vulnerability to poverty including low per capita income, high dependency on agricultural production, existence of risk factors, and poor infrastructure (see [12,13,39] for detailed explanation on sampling).

Next, a three-stage stratified random procedure was applied for the detailed sampling. In the first stage, sub-districts (in Thailand) and communes (in Vietnam) were chosen with regard to their proportion of the population in sampled districts. After that, in each sub-district/commune, two villages were selected with a probability proportional to their population size in the sub-district/commune. In the last stage, ten households were randomly chosen from a list of all households in each sampled village with equal probability of selection. As a result, a sample of 2,200 households from 220 villages in each country was identified for the first wave (or 4,400 households from 440 villages for both Thailand and Vietnam in total). At present, surveys under the TVSEP project fully conducted in all provinces of both Thailand and Vietnam include the waves of 2007, 2008, 2010, 2013, 2016, and 2017. For the wave of 2022, the TVSEP project could collect the data from households in all three provinces in Thailand and only two provinces in Vietnam (see https://www.tvsep.de/en/survey-documents for more information about the survey waves and survey instruments). Before the survey started, the local partners of the TVSEP project had obtained the ethical approval as required by local authorities. However, these documents are not available publicly to data users. The data of the TVSEP project are anonymous and the respondents were verbally asked for their consent to conduct the interview (see https://www.tvsep.de/en/survey-documents#c68719 for documentations of the data).

To construct the sample for this study, we first take the data from waves which yield equal time gaps for consistent analyses. Next, we also select those waves which have necessary data (for example, household's asset values are not available in the way of 2007). These criteria lead to the selection of three waves of 2010, 2013, and 2016. Because these waves have an equal time gap (every three years). Second, to better examine the effects of the COVID-19, the most recent wave conduced in 2022 should be included. Third, we use identical households (those households were interviewed in these selected waves) to have more accurate and reliable examination of the impacts. Last, we exclude households with missing data. These processes result in a sample of 2,517 households (1,386 households from Thailand and 1,131 households from Vietnam) for four years (2010, 2013, 2016, and 2022). Our reduced sample has an attrition rate of about 5.1% per wave (compared to the number of successful interviews in the first wave of 2007 for three provinces in Thailand and two provinces in Vietnam). Hence, the final sample consists of 10,068 observations from Thailand and Vietnam in four waves.

Besides the TVSEP data, we use two variables at provincial levels, namely the unemployment rate and share of rural population representing socio-economic conditions at the provincial levels. The data were collected from [40] for Thailand and from [41] for Vietnam. To a certain extent, these variables allow us to control for observable time-variant characteristics of the provinces where rural households in our sample live.

## Description of data

Table 1 stacks the descriptive summary of household characteristics in 2010, 2016, and 2022 (the descriptive statistics of 2013 is excluded for the sake of simplicity and to have an equal six-year gap of data; see S1 Table in the supplementary information for the detailed definition and measurement of variables). Panel A of this table shows the change of household's daily per capita income. It can be seen that the average daily per capita income of rural households in our sample is about PPP$ 7.10. The change of the income shows an increasing trend over time when it significantly increased from PPP$ 5.21 in 2010 to PPP$ 8.90 in 2016, and decreased to PPP$ 7.92 in 2022.

Panel B of Table 1 presents the descriptive statistics of important household characteristics. It appears that there were about 32% of rural households in our sample having at least a member contracted the COVID-19 in 2022. The average age of the household heads is about 58 years old, 75% of them are male, and about 87% of the heads belong to the group of ethnic majorities. In our sample, the average household size is about 4 persons with about 2 adults in each household. The average number of elderly increased from 0.53 to 0.90 between 2010 and 2022 implying a potential issue of aging population in rural areas. About a third of household heads have a membership in political and social organization (PSO). The share of farm laborers in total household laborers is about 71%, however, it shows a decreasing trend between 2010 and 2022 (from 76.46% in 2010 to 62.93% in 2022).

The average schooling years of household heads and adult members in households are about 5.97 years and 6.32 years, respectively. The average percentage of households reported a shock (e.g., any of weathers shock, health shocks, or economic shocks) in the past 12 months is around 55% in our sample. The average land area per capita of rural households in our sample is roughly 0.63 hectare (ha) and this figure was decreasing from 0.63 ha in 2010 to 0.53 ha in 2022. The average asset value per capita of households shows an increasing tendency between 2010 and 2022 (PPP$ 1,167 in 2010 vs. PPP$ 2,015 in 2022). In particular, the average asset value per capita increased from PPP$ 1,167 in 2010 to PPP$ 1,934 in 2016. The difference of the asset value between 2010 and 2016 was about PPP$ 767 and statistically significant. However, the difference of the asset value per capita was just PPP$ 81 between 2016 and 2022 (increased from PPP$ 1,934 in 2016 to PPP$ 2,015 in 2022) and it turned into statistically insignificant.

## Research method

In this section, we describe how we measure income inequality and poverty in the next two sub-sections. Then, we explain our empirical strategy to examine the correlation and heterogeneous effects of the COVID-19 on household

**Table 1. Descriptive summary of household characteristics.**

| | Whole sample (n = 10068) | By years | | | Mean difference/Statistical test | | |
|---|---|---|---|---|---|---|---|
| | | 2010 (n = 2517) | 2016 (n = 2517) | 2022 (n = 2517) | 2010 vs. 2016 | 2016 vs. 2022 | 2010 vs. 2022 |
| *A. Household income* | | | | | | | |
| Daily per capita income (PPP$) | 7.10 | 5.21 | 8.90 | 7.92 | 3.69***, a | −0.98**, a | 2.71***, a |
| | (14.28) | (6.88) | (15.44) | (15.39) | | | |
| *B. Household characteristics* | | | | | | | |
| A member contracted the COVID-19 (yes = 1) | 0.08 | 0.00 | 0.00 | 0.32 | 0.00b | 0.32***, b | 0.32***, b |
| | (0.27) | (0.00) | (0.00) | (0.46) | | | |
| Age of household head (years old) | 57.83 | 53.99 | 58.37 | 62.47 | 4.39***, a | 4.09***, a | 8.48***, a |
| | (12.75) | (12.84) | (12.09) | (11.94) | | | |
| Gender of head (male = 1) | 0.75 | 0.80 | 0.74 | 0.68 | −0.06***, b | −0.06***, b | −0.12***, b |
| | (0.43) | (0.40) | (0.44) | (0.47) | | | |
| Ethnicity of head (majority = 1) | 0.87 | 0.88 | 0.88 | 0.87 | 0.00b | −0.01b | −0.01b |
| | (0.33) | (0.33) | (0.33) | (0.34) | | | |
| Household size (persons) | 3.89 | 4.22 | 3.78 | 3.55 | −0.44***, a | −0.23***, a | −0.67***, a |
| | (1.71) | (1.71) | (1.65) | (1.73) | | | |
| Number of adults (persons) | 2.05 | 2.22 | 2.02 | 1.80 | −0.20***, a | −0.22***, a | −0.42***, a |
| | (1.27) | (1.18) | (1.26) | (1.33) | | | |
| Number of elderly members (persons) | 0.69 | 0.53 | 0.70 | 0.90 | 0.17***, a | 0.19***, a | 0.36***, a |
| | (0.81) | (0.76) | (0.81) | (0.82) | | | |
| Member of PSO (yes = 1) | 0.34 | 0.36 | 0.36 | 0.28 | 0.00b | −0.08***, b | −0.08***, b |
| | (0.47) | (0.48) | (0.48) | (0.45) | | | |
| Share of farm laborers (%) | 71.37 | 76.46 | 70.48 | 62.93 | −5.98***, a | −7.55***, a | −13.53***, a |
| | (32.80) | (30.02) | (31.69) | (36.58) | | | |
| Schooling years of household head (years) | 5.97 | 5.94 | 6.06 | 5.90 | 0.11a | −0.16*, a | −0.04, a |
| | (3.41) | (3.40) | (3.38) | (3.39) | | | |
| Mean schooling years of adult members (years) | 6.32 | 6.37 | 5.44 | 7.72 | −0.92***, a | 2.27***, a | 1.35***, a |
| | (2.83) | (2.44) | (2.78) | (2.88) | | | |
| Experienced a shock in the past 12 months (yes = 1) | 0.55 | 0.62 | 0.64 | 0.34 | 0.02b | −0.30***, b | −0.28***, b |
| | (0.50) | (0.49) | (0.48) | (0.47) | | | |
| Land area per capita (ha) | 0.63 | 0.63 | 0.62 | 0.53 | −0.01a | −0.09***, a | −0.10***, a |
| | (0.86) | (0.78) | (0.79) | (0.69) | | | |
| Asset value per capita (PPP$) | 1706.75 | 1166.79 | 1934.08 | 2014.65 | 767.29***, a | 80.58a | 847.86***, a |
| | (3676.94) | (2192.99) | (3636.76) | (4635.82) | | | |
| *C. Provincial socio-economic indicators* | | | | | | | |
| Unemployment rate (%) | 1.80 | 1.96 | 1.48 | 2.55 | −0.48***, a | 1.07***, a | 0.59***, a |
| | (1.38) | (1.09) | (1.01) | (1.78) | | | |
| Share of rural population (%) | 78.91 | 77.14 | 79.69 | 78.54 | 2.55***, a | −1.15***, a | 1.40***, a |
| | (4.01) | (4.75) | (3.41) | (3.42) | | | |

Note: Standard deviations in parentheses; a: Two-sample t-test; b: Non-parametric rank-sum test; *** $p < 0.01$, ** $p < 0.05$, * $p < 0.1$.

income, inequality, and poverty in the third sub-section. Last, we articulate our method to investigate the distributional effects of the COVID-19 on household income in the last sub-section.

## Measurement of income inequality

We use two approaches, namely the Lorenz curve and Gini coefficient, to measure income inequality [42]. We follow the construction of the Lorenz curves and calculation of Gini coefficient from [43]. In particular, the Gini coefficient of province $p$ ($G_p$) is calculated as follows:

$$G_p = 1 - \sum_{z=1}^{n} (X_z - X_{z-1})(Y_z + Y_{z-1})$$

(1)

where $X_z$, indexed in increasing order ($X_{z-1} < X_z$), is the cumulated proportion of the population ($z = 0, \ldots, n$, with $X_0 = 0$, $X_n = 1$) and $Y_k$ is the cumulative proportion of daily income per capita ($z = 0, \ldots, n$, with $Y_0 = 0$, $Y_n = 1$). $Y_z$ is indexed in non-decreasing order ($Y_z > Y_{z-1}$). Since each province might have different geographical conditions and the COVID-19 measures which could affect the income of local households, we construct the Lorenz curve and calculate the Gini coefficient of income inequality for each province in each year.

Fig 1 presents the changes of the Lorenz curves of daily per capita income in five provinces in 2016 and 2022 (see S1 Fig in the supplementary information for the Lorenz curves of daily per capita income in 2010 and 2013). It can be seen that there is an increasing inequality of daily per capita income between 2016 (before the COVID-19) and 2022 (after the COVID-19), especially in Buriram and Nakhon Phanom in Thailand.

Table 2 shows the descriptive summaries of the Gini coefficients of daily per capita income by provinces in 2010, 2016, and 2022. Overall, the Gini coefficient of household's daily per capita income is about 0.52 in our sample and there is a considerable increase of the coefficient after the COVID-19 (from 0.48 in 2016 to 0.54 in 2022). It appears that the Gini coefficients have increased between 2016 and 2022, but there are heterogeneous changes in the Gini coefficients across provinces. In Ubon Ratchathani, the increase of the Gini coefficient is just about 0.01 (from 0.51 in 2016 to 0.52 in 2022), while the changes of the Gini coefficients of Buriram and Nakhon Phanom are about 0.07 and 0.13, respectively (Buriram:

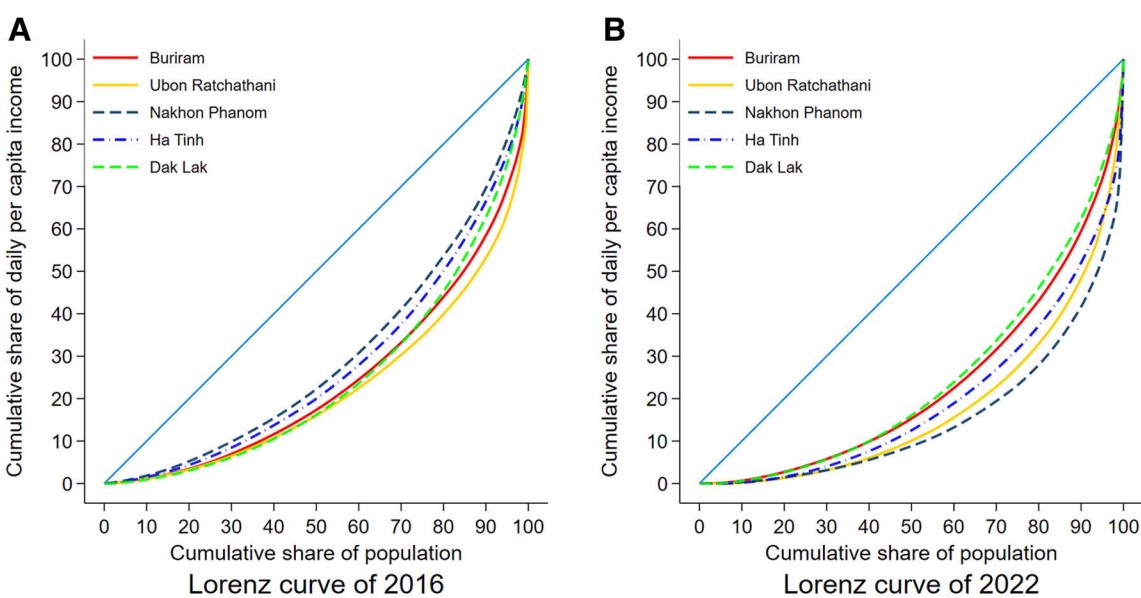

**Fig 1. Lorenz curves of daily per capita income in each province in 2016 and 2022.**

**Table 2. Descriptive summary of the Gini coefficients of daily per capita income by provinces and years.**

| | Number of observations | Overall | By years | | | Mean difference | | |
|---|---|---|---|---|---|---|---|---|
| | | | 2010 | 2016 | 2022 | 2010 vs. 2016 | 2016 vs. 2022 | 2010 vs. 2022 |
| Gini coefficient of income for each province in each year | 10068 | 0.52 | 0.49 | 0.48 | 0.54 | −0.01 | 0.06 | 0.05 |
| | | (0.06) | (0.02) | (0.05) | (0.06) | | | |
| *By provinces* | | | | | | | | |
| Buriram (Thailand) | 2,108 | 0.49 | 0.48 | 0.41 | 0.48 | −0.07 | 0.07 | 0.00 |
| | | (0.06) | (0.00) | (0.00) | (0.00) | | | |
| Ubon Ratchathani (Thailand) | 2,488 | 0.53 | 0.51 | 0.52 | 0.52 | 0.01 | 0.01 | 0.01 |
| | | (0.03) | (0.00) | (0.00) | (0.00) | | | |
| Nakhon Phanom (Thailand) | 948 | 0.59 | 0.46 | 0.55 | 0.68 | 0.09 | 0.13 | 0.22 |
| | | (0.09) | (0.00) | (0.00) | (0.00) | | | |
| Ha Tinh (Vietnam) | 2,260 | 0.51 | 0.50 | 0.45 | 0.58 | −0.05 | 0.13 | 0.08 |
| | | (0.05) | (0.00) | (0.00) | (0.00) | | | |
| Dak Lak (Vietnam) | 2,264 | 0.50 | 0.48 | 0.50 | 0.50 | 0.03 | −0.00 | 0.02 |
| | | (0.01) | (0.00) | (0.00) | (0.00) | | | |

Note: Standard deviations in parentheses.

from 0.41 in 2016 to 0.48 in 2022; Nakhon Phanom: from 0.55 in 2016 to 0.68 in 2022). The similar heterogeneity of the changes can also be noticed in two provinces of Vietnam. The Gini coefficient of Dak Lak remains almost unchanged at 0.50 in the period of 2016–2022 and the Gini coefficient of Ha Tinh surges from 0.45 to 0.58 in the same period.

## Measurement of poverty

In this study, we use two different approaches to the poverty measurement of rural households, namely absolute income poverty and multidimensional poverty. The key difference between these approaches is that the former classifies poverty at a fixed threshold of, for example, household income, while the latter takes into account several aspects (i.e., multidimensional aspects) of households such as income, education, and access to basic infrastructure [44]. Regarding absolute income poverty, we use the World Bank's poverty threshold for middle-income countries at a daily per capita income of PPP$ 3.20 [45] since Thailand and Vietnam are among countries in the middle-income group and our data fit best with the proposed period. In other words, a household is classified as living under absolute income poverty if their daily per capita income is at/lower than PPP$ 3.20. For the multidimensional poverty, we apply a measure suggested by the World Bank [44]. In this framework, besides the income poverty at PPP$ 3.20 as a monetary dimension, two additional dimensions of household conditions are added to classify whether a household is living under multidimensional poverty, namely education and access to basic infrastructure (Detailed definition and measurement of multidimensional poverty's indicators are presented in Panel C of S1 Table). Each dimension is equally weighted (the adopted measures, parameters, and weights are shown in S2 Table in the supplementary information). We set a cut-off level at one-third. In other words, a household is classified as living under multidimensional poverty if the total parameters of multidimensional poverty dimensions in that household is at least one-third (i.e., 0.333).

Table 3 shows the descriptive summary of the parameters for measuring multidimensional poverty and poverty indicators. The parameters of multidimensional poverty are presented in Panel A of Table 3. It can be seen that, in our sample, the share of households having no schooling of school-age children is only 5% in 2010 and reduces to 2% in 2016, but it increases to 6% in 2022. In a similar trend, the percentages of households with no primary education of adult members and households having no improved sanitation also show a decreasing tendency between 2010 and 2016 and an increasing tendency between 2016 and 2022. This might imply the impacts of COVID-19 on rural households, especially on

education. On the other hand, the share of households with unsafe sources of drinking water decreases from 44% in 2010 to 37% in 2016, and falls to a low at 29% in 2022. The percentage of households with no access to electricity remains nearly unchanged at 2% between 2010 and 2022.

Panel B of Table 3 stacks the indicators of absolute and multidimensional poverty. It can be seen that poverty of rural households is slightly more severe when we consider more dimensions of household education and access to basic infrastructure, as about 41% of rural households in our sample live in multidimensional poverty. Both the indicators of absolute and multidimensional poverty show a downward trend between 2010 and 2016 and an upward trend between 2016 and 2022.

## Investigating the correlation and the heterogenous effects of the COVID-19 on household income, inequality, and poverty

We start our empirical strategy by investigating the correlation of the COVID-19 pandemic on household income, income inequality, and poverty. Many studies used reported changes of livelihood, income, or expenditure as outcome variables to examine the effects of the COVID-19 [1,19,25–27,46]. This approach has, more likely, to rely on cross-sectional data and there is no actual variable representing the COVID-19 in their empirical model. On the other side, some studies used variables as proxies of the COVID-19 such as confirmed cases of the COVID-19 and post-lockdown/post-COVID-19 variables [22,34,37,47]. The former method can partly capture the impacts of the COVID-19 on the health of infected people in rural households and their livelihood due to a short-term isolation period, while the indicators of post-lockdown or post-COVID-19 period can reflect better how the COVID-19 actually affects the whole households. It is important to note that the TVSEP data collected in 2022 cover the worst infection wave of the COVID-19 with the variant B.1.1.7 (Alpha) in Thailand and B.1.617.2 (Delta) in Vietnam [31] and also lockdowns in these two countries [32,33]. Further, the COVID-19 pandemic should be considered as an aggregate shock which could affect every aspect of household living and livelihood

Table 3. Descriptive summary of multidimensional poverty parameters and poverty indicators.

| | Whole sample (n = 10068) | By years | | | Mean difference/Statistical test | | |
|---|---|---|---|---|---|---|---|
| | | 2010 (n = 2517) | 2016 (n = 2517) | 2022 (n = 2517) | 2010 vs. 2016 | 2016 vs. 2022 | 2010 vs. 2022 |
| *A. Parameters of multidimensional poverty* | | | | | | | |
| No schooling of school-age children (yes = 1) | 0.06 | 0.05 | 0.02 | 0.06 | −0.02***, a | 0.03***, a | 0.01a |
| | (0.23) | (0.21) | (0.15) | (0.23) | | | |
| No primary education of adult members (yes = 1) | 0.03 | 0.03 | 0.01 | 0.07 | −0.02***, a | 0.06***, a | 0.04***, a |
| | (0.18) | (0.17) | (0.11) | (0.25) | | | |
| Unsafe drinking water (yes = 1) | 0.39 | 0.44 | 0.37 | 0.29 | −0.07***, a | −0.08***, a | −0.15***, a |
| | (0.49) | (0.50) | (0.48) | (0.46) | | | |
| No improved sanitation (yes = 1) | 0.33 | 0.35 | 0.23 | 0.41 | −0.12***, a | 0.18***, a | 0.06***, a |
| | (0.47) | (0.48) | (0.42) | (0.49) | | | |
| No access to electricity (yes = 1) | 0.02 | 0.02 | 0.02 | 0.02 | −0.00a | 0.00a | 0.00a |
| | (0.14) | (0.13) | (0.12) | (0.14) | | | |
| *B. Poverty indicators* | | | | | | | |
| Income poverty at PPP$ 3.20 per capita a day (yes = 1) | 0.40 | 0.48 | 0.26 | 0.40 | −0.22***, a | 0.14***, a | −0.08***, a |
| | (0.49) | (0.50) | (0.44) | (0.49) | | | |
| Multidimensional poverty (yes = 1) | 0.41 | 0.49 | 0.27 | 0.41 | −0.22***, a | 0.14***, a | −0.08***, a |
| | (0.49) | (0.50) | (0.44) | (0.49) | | | |

Note: Standard deviations in parentheses; a: Non-parametric rank-sum test; *** p < 0.01.

conditions [48]. Therefore, a variable representing the COVID-19 period can be relevant to examine the influence of the COVID-19 [34]. We use a fixed-effects model to take the advantage of panel data to control for unobservable and time-invariant heterogeneities of rural households. The household fixed-effects model can be specified as follows:

$$Y_{ipw} = \beta_0 + \beta_1 COVID_w + \beta_2' X_{ipw} + \beta_3' P_{pw} + h_{ip} + \omega_{ipw} \tag{2}$$

where $Y_{ipw}$ refers to indicators of household $i$ from province $p$ at wave $w$ including (i) daily per capita income, (ii) Gini coefficient of income inequality, (iii) absolute income poverty, and (iv) multidimensional poverty. $COVID_w$ is a binary variable (COVID-19 period (i.e., wave of 2022) = 1; otherwise = 0). $X_{ipw}$ is a group of household variables. In this study, we control for important indicators of household livelihood which have been widely used in livelihood and poverty-related studies [12,49], namely age of head, gender of head, household size, number of adults in household, number of elderly members in household (as household's demographic characteristics), ethnicity of head, membership in political and social organization (as household's social capital), share of farm laborers, schooling years of head, average schooling years of adult members (as household's human capital), land area, and asset-poor household (as household's wealthy level). We further control for a member contracted the COVID-19 and exposure to any shocks in the past 12 months since these events might influence household income. $P_{pw}$ are province level's variables, namely unemployment rate and the share of rural population. $h_{ip}$ is the household fixed-effects and $\omega_{ipw}$ is the unobserved time-varying idiosyncratic errors.

In the next step, to evaluate the heterogeneous effects of the COVID-19, we include interaction terms between the COVID-19 variable ($COVID_w$) and three variables of household characteristics ($H_{ipw}$), namely gender of head (i.e., male head vs. female head), share of farm laborers (i.e., the extent of household's participation in farming), and schooling years of adult members (i.e., household's educational level). Again, we employ a fixed-effects model to estimate the heterogeneous impacts of the COVID-19 which can be written as follows:

$$Y_{ipw} = \gamma_0 + \gamma_1 COVID_w + \gamma_2 COVID_w * H_{ipw} + \gamma_3' X_{ipw} + \gamma_4' P_{pw} + h_{ip} + \varepsilon_{ipw} \tag{3}$$

In Equation 3, $Y_{ipw}$ is the group of dependent variables, namely household income, income inequality, and poverty variables, $COVID_w$ is the binary variable of the COVID-19 period, and $X_{ipw}$ and $P_{pw}$ are the groups of household and province variables which remain unchanged as explained in Equation 2. $H_{ipw}$ is the group of three household variables which are used to examine the heterogeneous effects of the COVID-19 ($H_{ipw}$ belongs to $X_{ipw}$). $h_{ip}$ is the household fixed-effects and $\varepsilon_{ipw}$ is the unobserved time-varying idiosyncratic errors. We check for a problem of multicollinearity of independent variables in Equations 2 and 3. The mean VIF values show no evidence of this problem (see Column 1 of S3 Table in the supplementary information for the model without interaction terms and Columns 2, 3, and 4 for the models with interaction terms). We cluster these estimates at the village level to have robust standard errors.

## Examining the distributional effects of the COVID-19 on household income

In the last step, we complete our empirical strategy by evaluating how the COVID-19 affects rural households in different income groups. We use an unconditional quantile regression (UQR) model suggested by [50] to examine the distributional effects of the COVID-19 on household income. The key advantage of this UQR approach is that it can estimate unconditional partial effects (UPE) of changes in the distribution of independent variables on the distributional statistic of an outcome variable through a two-step procedure and, at the same time, it can include robust and clustered standard errors [51–53]. In the first step, this UQR model calculates the re-centered influence function (RIF) as:

$$RIF(Y; q_\tau, F_P) = q_\tau + \frac{\tau - 1\{Y \le q_\tau\}}{f_Y(q_\tau)} \tag{4}$$

where $q_\tau$ is the value of household' daily per capita income, $Y$, at the quantile $\tau$. $F_P$ denotes the cumulative distribution function of the outcome variable $Y$, and $f_Y(q_\tau)$ is the density of $Y$ at $q_\tau$. $1\left\{Y \leq q_\tau\right\}$ is an indicator function to identify whether the value of outcome variable $Y$ is below $q_\tau$.

In the second step, this model estimates the impacts of the COVID-19 on household's income ($Y$) as follows:

$$I\left[RIF\left(Y_{ip}; q_\tau\right)|X, COVID\right] = \delta + \eta COVID + \vartheta X_{ip} + \psi P_p + \in_{ip} \tag{5}$$

where $Y_{ip}$ is either daily per capita income of household $i$ from province $p$. **COVID** is the binary variable of the COVID-19 period and $X_{ip}$ and $P_p$ are the groups of household and province variables as defined in Equations 2 and 3, respectively. The UPE of the COVID-19 can be calculated as $\eta = \frac{\partial v(F_P)}{\partial \overline{R}_k}$ [53]. We cluster the estimation of UQR models at the village level to have robust standard errors.

## Results and discussion

### The correlation and the heterogenous effects of the COVID-19 on household's income, inequality, and poverty

Table 4 presents the results of the COVID-19's correlation with household's daily per capita income, Gini coefficient of income inequality, absolute income poverty, and multidimensional poverty. It appears that the COVID-19 has a negative correlation with household income. In particular, the coefficients of the COVID-19 variable show that, compared to the before-the-COVID-19 period, the COVID-19 has resulted in a decrease of the daily per capita income of rural households by 19.67% (since the COVID-19 variable is a dummy, the semi-elasticity is calculated as "100*(EXP(coefficient)-1)%"). The negative correlation of the COVID-19 with household income is consistent with the findings from [17,18,22,46].

The inevitable consequences of this reduction in income are the raising inequality and poverty. Our results show that the COVID-19 has a positive correlation with the Gini coefficient of income, absolute income poverty, and multidimensional poverty. The coefficients of the COVID-19 variable point to the finding that the COVID-19 increases the Gini coefficient by 0.013 and the incidences of absolute poverty and multidimensional poverty by 7.1% compared to the before-the-COVID-19 period. Our results of the influence of the COVID-19 on inequality and poverty are in the same vein as those findings from [2,20,22,25], and [47].

We further find that the variable of household having at least a member contracted the COVID-19 has a positive and significant correlation with the Gini coefficient of income inequality. This shows a direct influence of the COVID-19 on income inequality at the provincial level. The age of head, numbers of adults and elderly members in the household, schooling years of head and mean schooling years of adult members have positive correlations with household's daily per capita income, while household size, share of farm laborers, shock exposure, and asset-poor household have negative correlations with the household's per capita income. The two variables at provincial level, namely unemployment rate and the share of rural population, also have negative correlations with the per capita income of households. Regarding the Gini coefficient, besides the variable of households with at least a member contracted the COVID-19, we find that male-headed households, household size, share of farm laborers, mean schooling years of adult members, land area per capita, province's unemployment rate and share of rural population have positive correlations with the Gini coefficient, while number of elderly members, education of heads, and shock exposure have negative correlations with the Gini coefficient. In addition, the age of head, number of adults, number of elderly members, schooling years of head, average schooling years of adult members, and land area per capita have negative associations, while household size, share of farm laborers, asset-poor households, province's unemployment rate and the share of rural population have positive associations with absolute and multidimensional poverty.

Table 5 shows the brief results of the heterogeneous effects of the COVID-19 on household's daily per capita income, Gini coefficient of income inequality, absolute income poverty, and multidimensional poverty (see S4 Table, S5 Table, and S6 Table in the supplementary information for full results). It can be seen that, in the context of the

**Table 4. Correlation of the COVID-19 with household income, Gini coefficient of income, and poverty indicators (Fixed-effects estimations).**

| | Daily per capita income (ln) | Gini coefficient of household income | Income poverty at PPP$ 3.20† | Multidimensional poverty† |
|---|---|---|---|---|
| COVID-19 period† | −0.219*** | 0.013*** | 0.071*** | 0.071*** |
| | (0.073) | (0.003) | (0.018) | (0.017) |
| Member contracted the COVID-19† | −0.082 | 0.007*** | −0.013 | −0.018 |
| | (0.092) | (0.002) | (0.022) | (0.022) |
| Age of head | 0.018*** | 0.000 | −0.004*** | −0.004*** |
| | (0.004) | (0.000) | (0.001) | (0.001) |
| Male head† | −0.138 | 0.009*** | 0.014 | 0.017 |
| | (0.089) | (0.003) | (0.023) | (0.023) |
| Ethnic majority† | 0.104 | −0.007 | −0.028 | −0.029 |
| | (0.180) | (0.006) | (0.045) | (0.045) |
| Household size | −0.236*** | 0.001* | 0.090*** | 0.090*** |
| | (0.026) | (0.001) | (0.007) | (0.007) |
| Number of adults | 0.218*** | −0.000 | −0.092*** | −0.092*** |
| | (0.036) | (0.001) | (0.009) | (0.009) |
| Number of elderly members | 0.176*** | −0.004** | −0.087*** | −0.087*** |
| | (0.055) | (0.002) | (0.014) | (0.014) |
| PSO member† | −0.025 | −0.000 | 0.005 | 0.011 |
| | (0.074) | (0.002) | (0.015) | (0.015) |
| Share of farm laborers | −0.004*** | 0.000*** | 0.001*** | 0.001*** |
| | (0.001) | (0.000) | (0.000) | (0.000) |
| Schooling years of head | 0.035*** | −0.003*** | −0.006* | −0.006* |
| | (0.013) | (0.000) | (0.003) | (0.003) |
| Mean schooling years of adult members | 0.036*** | 0.002*** | −0.012*** | −0.012*** |
| | (0.010) | (0.000) | (0.002) | (0.002) |
| Shock exposure† | −0.085** | −0.010*** | 0.010 | 0.010 |
| | (0.043) | (0.002) | (0.011) | (0.011) |
| Land area per capita | 0.012 | 0.013*** | −0.032*** | −0.032*** |
| | (0.058) | (0.001) | (0.009) | (0.009) |
| Asset poor† | −0.238*** | 0.002 | 0.090*** | 0.088*** |
| | (0.057) | (0.002) | (0.014) | (0.014) |
| Province's unemployment rate | −0.095*** | 0.012*** | 0.023*** | 0.022*** |
| | (0.025) | (0.001) | (0.006) | (0.006) |
| Province's share of rural population | −0.022*** | 0.006*** | 0.005** | 0.005** |
| | (0.008) | (0.001) | (0.002) | (0.002) |
| Constant | 2.383*** | 0.025 | 0.111 | 0.102 |
| | (0.682) | (0.052) | (0.194) | (0.192) |
| Number of observations | 10068 | 10068 | 10068 | 10068 |
| F (17,361) | 15.485 | 25.436 | 26.474 | 25.799 |
| Prob. > F | 0.000 | 0.000 | 0.000 | 0.000 |
| R-squared: | | | | |
| Within | 0.031 | 0.182 | 0.053 | 0.053 |
| Between | 0.085 | 0.390 | 0.154 | 0.172 |
| Overall | 0.048 | 0.233 | 0.091 | 0.099 |

Note: Robust standard errors clustered at village level in parentheses; †: Dummy; ln: natural logarithm; *** $p < 0.01$, ** $p < 0.05$, * $p < 0.1$.

**Table 5.** Heterogeneous effects of the COVID-19 on household income, Gini coefficient of household income, and poverty (Fixed-effects estimations).

| | Daily per capita income (ln) | Gini coefficient of household income | Income poverty at PPP$ 3.20† | Multidimensional poverty† |
|---|---|---|---|---|
| *A. Heterogeneous effect by the gender of head* | | | | |
| COVID-19*Male head | 0.056 | 0.005** | −0.045* | −0.049** |
| | (0.089) | (0.002) | (0.024) | (0.024) |
| *B. Heterogeneous effect by the share of farm laborers* | | | | |
| COVID-19*Share of farm laborers | −0.004*** | −0.000** | 0.001 | 0.000 |
| | (0.001) | (0.000) | (0.000) | (0.000) |
| *C. Heterogeneous effect by the education of adult members* | | | | |
| COVID-19*Mean schooling years of adults | 0.027* | 0.003*** | −0.018*** | −0.018*** |
| | (0.016) | (0.000) | (0.004) | (0.004) |

Note: Robust standard errors clustered at village level in parentheses; †: Dummy; ln: natural logarithm; *** $p<0.01$, ** $p<0.05$, * $p<0.1$. Full results presented in S4 Table, S5 Table, and S6 Table in the supplementary information.

COVID-19, households with male heads have positive effects on the Gini coefficient of income inequality and negative effects on poverty variables. The effects of the COVID-19 on females appear to be much worse than male counterparts [24,25,27,54,55]. The interaction between the COVID-19 variable with the share of farm laborers shows negative effects on household's daily per capita income and the Gini coefficient of income inequality. This finding indicates that farming households are more vulnerable under the impacts of the COVID-19 [35,37]. In contrast, in the context of the COVID-19, average schooling years of adult members have positive effects on household's daily per capita income and the Gini coefficient of income inequality, but it has a negative impact on absolute income poverty and multidimensional poverty. This result implies the important role of education in rural regions [34,47,49,54]. Overall, these findings from our study confirm the unequal effects of the COVID-19 on household income, income inequality, and poverty at the micro level [1,3,4].

## The distributional effects of the COVID-19 on household income

Table 6 stacks the results of how the effects of the COVID-19 are distributed across five quantiles of household's daily per capita income. It can be seen that the COVID-19 has negative effects on the daily per capita income, however, the impacts are unequal under different quantiles. In particular, the COVID-19 has a negative effect on income of households in the 10th, 25th, and 50th quantiles. The magnitude of the negative impact is highest in the 10th quantile group at around 44.8%. At the 25th income quantile, the negative impact of the COVID-19 is slightly smaller, but the reduction is

**Table 6.** Distributional effects of the COVID-19 on household income.

| | Income quantiles | | | | |
|---|---|---|---|---|---|
| | 10th | 25th | 50th | 75th | 90th |
| COVID-19 period† | −0.385*** | −0.448*** | −0.487*** | −0.122 | −0.305 |
| | (0.096) | (0.106) | (0.158) | (0.300) | (0.725) |
| Sample mean RIF | 0.859 | 2.000 | 4.202 | 8.007 | 14.341 |
| Impact magnitude on household's daily per capita income | −44.82% | −22.40% | −11.59% | Not significant | Not significant |

Note: Robust standard errors clustered at village level in parentheses; †: Dummy; *** $p<0.01$. Full results presented in S7 Table in the supplementary information.

still relatively high at more than 22.4%. At the 50th quantile group, the reduction of income is 11.6%. On the other hand, the negative effects of the COVID-19 on daily per capita income of households in the 75th and 90th quantile groups are not statistically significant. This explains why there is an increase in the Gini coefficient of income inequality in the COVID-19 period. Our results are supported by the findings from [16] and [56] that wealthier households might not experience a large reduction in income from wage- and self-employment activities compared to agricultural farmers. The rich get richer and the poor get poorer as a consequence of the COVID-19 [6].

As a robustness check, we employ the relative distribution approach from [57] to estimate the balancing weights of the covariates before and after the COVID-19 (see [58] for further explanation on the approach). We use the "reldist" command from STATA with "pdf" as the relative density and the entropy balancing method to generate the balancing weights. These weights are, then, applied to the estimations of UQR models. The results of these estimations on the distributional effects on household income using balancing weights are presented in S8 Table in the supplementary information. Overall, the negative effects of the COVID-19 on household income can still be observed. One difference is that the negative effects on the income of households in the 90th quantile group are now significant. However, the largest effects are still on the income of households in the 10th and 25th quantile groups with the reduction of 111% and 56%, respectively. The negative effects of the COVID-19 on household income's distribution remain consistent after applying the balancing weights.

## Conclusions and policy implications

The introduction of physical distancing measures (i.e., lock down) and the disruptions in production and supply chains caused by the COVID-19 pandemic have resulted in reduced livelihoods, increased food insecurity, increased poverty, and increased inequality. Given the limited evidence on the impacts of the COVID-19 at the micro level, we use panel data for Thailand and Vietnam to examine (i) the correlation and heterogeneous effects of the COVID-19 pandemic on household income, income inequality, and poverty and (ii) the distributional effects of the pandemic on household income. The results of fixed-effects estimations show that the COVID-19 has a negative correlation with household income, while it has a positive correlation with the Gini coefficient of income inequality, absolute poverty, and multidimensional poverty. In particular, compared to the before-the-COVID-19 period, the COVID-19 has resulted in a decrease of the daily per capita income of rural households by 19.67%. The COVID-19 increases the Gini coefficient of income inequality by 0.013 and the percentages of household living under absolute poverty and multidimensional poverty by 7.1% compared to the before-the-COVID-19 period.

The estimation results of the fixed-effects models with interaction terms between the COVID-19 variable and household characteristics point to important findings of the heterogeneous effects of the COVID-19. First, the effects of the COVID-19 on females appear to be much worse than male counterparts in terms of income inequality. Second, the interaction between the COVID-19 variable with the share of farm laborers shows negative effects on household's daily per capita income implying that farming households are more vulnerable under the impacts of the COVID-19. Last, average schooling years of adult members have positive effects on household's daily per capita income, and, at the same time, the average schooling years have positive impacts on absolute income poverty and multidimensional poverty. For the distributional effects of the COVID-19, the results of unconditional quantile regression models show that the impacts of the COVID-19 have the highest magnitude on daily per capita income of households in the 10th and 25th quantile groups, while the negative effects on daily per capita income in the 75th and 90th quantile groups are not statistically significant.

Relied on the above findings, several policy implications are proposed. First, governments in developing countries should put more emphases on supporting poor households and disadvantageous groups, especially females. People in these groups are vulnerable even before the outbreak of the COVID-19, the pandemic has widened the disparity between disadvantageous and better-off groups in terms of economic prosperity, resources, and access to alternatives to cope with unexpected events. In the short term, social safety net policies would be suited to coping with income decline, poverty, and food insecurity [3,23,59]. In the longer term, creating employment opportunities for rural households would be strongly

relevant to increase their livelihoods, for instance, by improving rural infrastructure to provide rural households with opportunities to participate in better income generation activities [12,60,61]. Second, the role of education remains extremely important to help rural households deal with the COVID-19. Therefore, promoting education in rural areas should be implemented. Last, agriculture is crucial in rural regions of developing countries to provide livelihoods and to ensure food security, however, agricultural farmers are also more vulnerable to external shocks such as the COVID-19. In addition, physical distancing measures caused by the pandemic have driven thousands of migrated workers in developing countries back to their rural villages which emphasizes the importance of having better rural economies. Hence, policies on stimulating agricultural transformation would be recommended to enable rural villages to engage in non-farm employment which can bring higher profit and improve their welfare [14,62].

Although our study has provided some useful insights, it is not free from limitations. First, we use the panel data which have inconsistent time gaps between panel waves. There was a large time gap between panel waves which might not well reflect the short-term change of household's livelihood before, during, and after the COVID-19. Hence, the use of panels with shorter time gaps is strongly recommended in future studies. Second, although the attrition rate of our sample is low, due to the fact that one province in Vietnam was excluded in the 2022 survey, the results should be interpreted strictly for the reduced sample. Third, the Gini coefficients of income inequality in our model are calculated at the provincial levels, hence the results regarding the correlation of the COVID-19 with the Gini coefficient should be interpreted with caution. Last, even though the impacts of the COVID-19 are enormous and we have tried to control for confounding factors happened in 2022, we might not be able to address these issues completely. Thus, the results should be interpreted as correlations, rather than causal effects.

## Supporting information

**S1 Fig. Lorenz curves of daily per capita income in each province in 2010 and 2013.**
(PDF)

**S1 Table. Name, definition and measurement of variables.**
(PDF)

**S2 Table. The adopted and adjusted measure of multidimensional poverty.**
(PDF)

**S3 Table. Variance inflation factor (VIF) values of independent variables.**
(PDF)

**S4 Table. Heterogeneous effects of the COVID-19 on household income, Gini coefficient, and poverty (Fixed-effects estimations): The case of head gender.**
(PDF)

**S5 Table. Heterogeneous effects of the COVID-19 on household income, Gini coefficient, and poverty (Fixed-effects estimations): The case of the share of farm laborers.**
(PDF)

**S6 Table. Heterogeneous effects of the COVID-19 on household income, Gini coefficient, and poverty (Fixed-effects estimations): The case of the education of adult members.**
(PDF)

**S7 Table. Distributional effects of the COVID-19 on household income.**
(PDF)

**S8 Table: Robustness check of the distributional effects of the COVID-19 on household income using balancing weights.**
(PDF)

**S1 Data. Replications.**
(ZIP)

## Acknowledgments

This study relies on the data from the long-term project No. 20220831434900116103. For more detailed information about the data, see www.tvsep.de. The authors would like to thank the respondents from the surveyed provinces in Thailand and Vietnam for their kind support and cooperation. We highly appreciate the effort of our colleagues at the Leibniz University Hannover for data collection and cleaning. Technical support from Mr. Kasem Kunasri is appreciated. We greatly acknowledge the helpful and constructive comments and suggestions from the Editor and two anonymous reviewers.

## Author contributions

**Conceptualization:** Manh Hung Do, Trung Thanh Nguyen.

**Formal analysis:** Manh Hung Do.

**Methodology:** Manh Hung Do, Trung Thanh Nguyen.

**Supervision:** Trung Thanh Nguyen, Ulrike Grote.

**Writing – original draft:** Manh Hung Do.

**Writing – review & editing:** Manh Hung Do, Trung Thanh Nguyen, Ulrike Grote.

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
