## [Decision Letter · Decision Letter 0]

25 Sep 2025

Dear Dr. Do,

Thank you for submitting your manuscript to PLOS ONE. After careful consideration, we feel that it has merit but does not fully meet PLOS ONE’s publication criteria as it currently stands. Therefore, we invite you to submit a revised version of the manuscript that addresses the points raised during the review process.

I have now received two reports on your paper. The referees make a number of constructive recommendations for improvement. While I am unable to accept the paper for publication in its current form, I would be pleased to reconsider it if you revise the manuscript carefully in line with the referees’ suggestions.

I should emphasize that some of the revisions requested are quite substantial. I hope you will be willing to undertake this additional work, as otherwise I will be unable to give further consideration to the paper.

We look forward to receiving your revised manuscript.

Kind regards,

Pablo Gutierrez Cubillos

Academic Editor

PLOS ONE

Journal Requirements:

“This study relies on data from the long-term project No. 20220831434900116103, funded by the Deutsche Forschungsgemeinschaft (DFG).”

4. Please include captions for your Supporting Information files at the end of your manuscript, and update any in-text citations to match accordingly. Please see our Supporting Information guidelines for more information: http://journals.plos.org/plosone/s/supporting-information .

Additional Editor Comments (if provided):

I have now received two reports on your paper. The referees make a number of constructive recommendations for improvement. While I am unable to accept the paper for publication in its current form, I would be pleased to reconsider it if you revise the manuscript carefully in line with the referees’ suggestions.

I should emphasize that some of the revisions requested are quite substantial. I hope you will be willing to undertake this additional work, as otherwise I will be unable to give further consideration to the paper.

Reviewers' comments:

Reviewer's Responses to Questions

**Comments to the Author**

1. Is the manuscript technically sound, and do the data support the conclusions?

Reviewer #1: Yes

Reviewer #2: Partly

2. Has the statistical analysis been performed appropriately and rigorously?

Reviewer #1: No

Reviewer #2: Yes

3. Have the authors made all data underlying the findings in their manuscript fully available?

Reviewer #1: Yes

Reviewer #2: Yes

4. Is the manuscript presented in an intelligible fashion and written in standard English?

Reviewer #1: Yes

Reviewer #2: Yes

Reviewer #1: The paper makes a valuable contribution by using robust panel data and appropriate econometric techniques to analyze the multifaceted impacts of COVID-19 on rural households in Thailand and Vietnam. The findings on increased poverty, inequality, and the heterogeneous/distributional effects are largely compelling. However, there are a few critical points, particularly concerning the Gini coefficient analysis and some discrepancies in reported results, that need attention.

1. Page 7, text description of asset value: "when it drops from PPP 769 between 2010 and 2016 to only PPP 78 between 2016 and 2022." This refers to the increase in asset value during those periods, not the asset value itself. Phrasing could be clearer to avoid misinterpretation. Moreover, this mean difference is not significant.

2. A member contracted to the COVID-19 (yes = 1)" variable (Table 1 & 4), Page 6 states "about 31% of rural households in our sample having a member contracted to the COVID-19 in 2022." This figure seems high for actual infection rates in a broad rural sample, even during peak waves. Could this variable be capturing households reporting a member contracted, or perhaps a broader definition of being directly affected by illness? Clarification on how this was measured would be beneficial.

3. In Table 4, this variable's coefficient for income is -0.092 but not statistically significant. Its effect on the Gini coefficient is negative and significant (-0.010), which is an interesting finding if the Gini is at the provincial level (discussed under Methodology).

4. Gini Coefficient (Table 2 & as dependent variable in Table 4):

A Gini coefficient of 0.90 for Nakhon Phanom in 2022 (Table 2) is extraordinarily high, bordering on perfect inequality. While possible, such an extreme value warrants a double-check of the calculation or a brief comment on potential drivers or data particularities for that province-year.

The more significant concern is its use as a household-level dependent variable (Y_iw) in Table 4. A Gini coefficient is an aggregate measure of inequality for a population (e.g., a province or village). It's not a household-level attribute.

If it's the province-level Gini: The unit of analysis for this dependent variable is province-wave. The regression in Table 4 lists N=10084 (household-wave observations). This implies each household within a province-wave is assigned the same Gini value. While this is sometimes done, it artificially inflates N for the Gini regression and standard errors need very careful handling (clustering at province level would be essential, village level might not be sufficient if Gini is province-level). The interpretation of household-level X_iw predicting a province-level Gini also becomes nuanced.

If it's some non-standard household-level Gini: This would require extensive justification and explanation, as it's not a conventional measure. This is the most critical methodological point to address.

5. Ensure Consistency in Reported Results: Correct the discrepancies in percentage/point changes between the main text summary (page 20) and the regression tables (Table 4).

6. Correct Interpretation of Education Interaction: Ensure the text on page 17 accurately reflects that education reduces the poverty impact of COVID-19, consistent with Table 5.

Reviewer #2: This paper examines the correlation between income, inequality, and poverty at the provincial level for Thailand and Vietnam, using panel data and focusing on the year 2022, which the authors treat as the COVID-19 year. The analysis relies on a fixed-effects model, followed by an exploration of heterogeneous effects. The main conclusion is that the year 2022 is positively correlated with the Gini coefficient and poverty, and negatively correlated with income.

While I appreciate the motivation of the paper and its objectives, I have concerns about the identification strategy. In particular, I am not convinced that treating 2022 as a proxy for COVID-19 captures the pure effect of the pandemic on socioeconomic variables. Other policies and shocks that occurred in 2022 may confound this proxy. To make the analysis cleaner, I suggest adding provincial-level controls such as unemployment and the percentage of households receiving social aid.

I also recommend including a dedicated limitations section. In this section, the authors could explicitly discuss the concerns raised above and acknowledge other caveats that may affect their results.

In addition, the robustness of the findings would be strengthened by applying the reweighting method of DiNardo, Fortin, and Lemieux (1996). In particular, the authors could reweight the provincial income distribution using observables from the 2016 wave as a counterfactual benchmark.

Finally, the paper would benefit from a careful proofreading. For example, in the introduction the authors write “… further deepened the inequality …”. Inequality of what? Clarifying such statements would improve readability. In terms of technical presentation, I suggest:

In equation (1), use the subindex w for provinces.

In equation (2), present definitions for indices i and w immediately after the equation.

With these improvements, the paper could make a meaningful contribution to the literature on the socioeconomic consequences of COVID-19.

**Do you want your identity to be public for this peer review?** For information about this choice, including consent withdrawal, please see our Privacy Policy

Reviewer #1: No

Reviewer #2: No

---

## [Author Response · Author response to Decision Letter 1]

25 Oct 2025

Our detailed response to each of the Editor and reviewers' comments is included in the "Response_to_editor_reviewers" letter.

---

## [Editor Report · Decision Letter 1]

21 Dec 2025

Dear Dr. Do,

Thank you for submitting your manuscript to PLOS ONE. After careful consideration, we feel that it has merit but does not fully meet PLOS ONE’s publication criteria as it currently stands. Therefore, we invite you to submit a revised version of the manuscript that addresses the points raised during the review process.

We look forward to receiving your revised manuscript.

Kind regards,

Pablo Gutierrez Cubillos

Academic Editor

PLOS One

Journal Requirements:

Additional Editor Comments (if provided):

Thank you very much for your revisions, which have significantly improved the paper.

However, I would appreciate it if the authors could provide a more in-depth response to the third comment from Referee 2, as their current answer lacks sufficient accuracy and clarity. Specifically, while both the DiNardo, Fortin, and Lemieux (1996) reweighting approach and Unconditional Quantile Regression (Firpo et al., 2009) are motivated by distributional analysis, they are distinct methods that address different objectives, and they are not equivalent.

The DFL approach constructs counterfactual income distributions by reweighting observables, while UQR estimates marginal effects on unconditional quantiles. Therefore, the claim that UQR is "similar" to DFL and renders reweighting redundant is not fully justified. Furthermore, it is entirely feasible to apply the DFL reweighting and then estimate UQR models on the reweighted sample.

I would therefore request that the authors either implement the suggested reweighting exercise as a robustness check or offer a clearer, technically sound explanation of why this approach might not be suitable for their specific empirical context. They should also explicitly address the conceptual differences between the two methods to avoid any confusion.

---

## [Author Response · Author response to Decision Letter 2]

8 Jan 2026

Our response to the Editor's comments can be found in the "2. Response_to_editor" file.

---

## [Editor Report · Decision Letter 2]

11 Jan 2026

Households’ poverty and inequality after the COVID-19: Insights from panel data of face-to-face surveys in Southeast Asia

PONE-D-25-22204R2

Dear Dr. Do,

We’re pleased to inform you that your manuscript has been judged scientifically suitable for publication and will be formally accepted for publication once it meets all outstanding technical requirements.

Kind regards,

Pablo Gutierrez Cubillos

Academic Editor

PLOS One
---

## [Editor Report · Acceptance letter]

PONE-D-25-22204R2

PLOS One

Dear Dr. Do,

I'm pleased to inform you that your manuscript has been deemed suitable for publication in PLOS One. Congratulations! Your manuscript is now being handed over to our production team.

Kind regards,

on behalf of

Dr. Pablo Gutierrez Cubillos

Academic Editor

PLOS One